# Lithium enrichment in intracontinental rhyolite magmas leads to Li deposits in caldera basins

Thomas R. Benson [1], Matthew A. Coble[1], James J. Rytuba[2] & Gail A. Mahood[1]

The omnipresence of lithium-ion batteries in mobile electronics, and hybrid and electric vehicles necessitates discovery of new lithium resources to meet rising demand and to diversify the global lithium supply chain. Here we demonstrate that lake sediments preserved within intracontinental rhyolitic calderas formed on eruption and weathering of lithium-enriched magmas have the potential to host large lithium clay deposits. We compare lithium concentrations of magmas formed in a variety of tectonic settings using in situ trace-element measurements of quartz-hosted melt inclusions to demonstrate that moderate to extreme lithium enrichment occurs in magmas that incorporate felsic continental crust. Cenozoic calderas in western North America and in other intracontinental settings that generated such magmas are promising new targets for lithium exploration because lithium leached from the eruptive products by meteoric and hydrothermal fluids becomes concentrated in clays within caldera lake sediments to potentially economically extractable levels.

[1] Department of Geological Sciences, Stanford University, 450 Serra Mall, Building 320, Stanford, CA 94305, USA. [2] United States Geological Survey, 345 Middlefield Road, Menlo Park, CA 94025, USA. Correspondence and requests for materials should be addressed to T.R.B. (email: trb@stanford.edu)

Recognition of the climatic impact of anthropogenic greenhouse gas emissions has led to the development of sustainable energy technologies requiring unconventional ore resources[1] identified as "critical" or "strategic" based on their importance to clean energy and the potential geopolitical risk to supply[2, 3]. Lithium (Li) is classified as an energy-critical element by several governments[4] due to increasing demand for Li-ion batteries, which have a high power density and relatively low cost that make them optimal for energy storage in portable electronic devices, the electrical power grid, and the growing fleet of hybrid and electric vehicles[5, 6]. Although current annual consumption of Li is small (~32.5 kt/a) compared to the estimated global economically extractable Li reserve of ~14 Mt[7], projections of future Li demand through 2050 range from ~3–35 Mt[5, 6, 8, 9] (Fig. 1), with the supply/demand balance likely becoming critical by 2030[10]. In addition, the current market share of Li is dominated by Australia and Chile, who together account for more than three-quarters of the world's Li production (Fig. 1)[7], highlighting the strategic necessity for governments and manufacturers to reduce import reliance by securing additional domestic Li resources.

Lithium resources, which represent the total amount of Li extractable under current (reserves) plus potentially foreseeable economic conditions, occur primarily as pegmatites, brines, and clays (Figs. 1 and 2a). Pegmatite Li deposits form during very late-stage crystallization of water-rich, rhyolitic magma, during which Li minerals lepidolite and spodumene crystallize[11, 12]. Due to their high grades and global distribution, pegmatite Li deposits account for approximately half of global Li production, the majority of which is produced from pegmatites in Australia (Figs. 1 and 2a)[7]. Approximately 35% of the current production comes from two saline brine deposits in salars in Chile (Figs. 1 and 2a)[7], which form on evaporation within closed basins of meteoric water that has leached Li from surficial rhyolitic rocks[11]. In clay deposits, Li is leached from rhyolitic lavas and volcanic ash by meteoric and hydrothermal fluids, and is structurally bound in clay (e.g., hectorite; $Na_{0.3}[Mg,Li]_3Si_4O_{10}[OH]_2$) developed in ash-rich sediments in basins adjacent to the source rocks[13, 14]. These deposits are gaining more attention due to the recent assessment of the McDermitt/Kings Valley deposit in Nevada (Fig. 2) as the largest Li resource in the United States (~2 Mt)[15], and the selection of the Li clay deposit in Sonora, Mexico (Fig. 2a), as the future supply for the Tesla Motors gigafactory in Reno, Nevada[16].

The association of all types of Li deposits with felsic rocks is a direct result of the incompatibility of Li in the structure of nearly all minerals that crystallize from rhyolitic melt (e.g., feldspar, quartz, and pyroxene)[17], resulting in enrichment of residual melts formed during extreme fractional crystallization[11, 18–21]. To elucidate the principle mechanisms responsible for enrichment of rhyolitic magmas in Li, we compare Li concentrations measured in situ on homogenized quartz-hosted melt inclusions from rhyolitic ignimbrites and lavas from different geological settings (Fig. 2; Supplementary Data 1) utilizing the SHRIMP-RG ion microprobe. We find that rhyolitic magmas that contain a significant proportion of felsic continental crustal material have elevated Li concentrations similar to those observed in rare-metal granites (>1000 ppm), whereas concentrations are an order of magnitude smaller in rhyolites erupted through thin, mafic continental crust. Although the largest Li deposits worldwide occur in brines within basins adjacent to rare aluminous rhyolites extremely enriched in Li (>2000 ppm)[5, 11], here, we show that more common, voluminous rhyolites only moderately enriched in Li can give rise to large Li clay deposits in lacustrine sediments within associated calderas.

## Results

**Magmas analyzed in this study.** Most of our analyzed samples are from the Middle Miocene McDermitt volcanic field (MVF) of Oregon and Nevada (Fig. 2b), where the largest Li resource in the United States, the Kings Valley deposit, is hosted within caldera lake sediments[10, 13, 15]. Four peralkaline (molar $Na_2O+K_2O/Al_2O_3 > 1$) ignimbrites erupted at MVF as a result of intrusion of dike swarms of Columbia river flood basalt associated with impingement of the Yellowstone plume head into transitional continental crust at the western margin of the North American craton[22, 23]: 16.47 Ma Tuff of Oregon Canyon, 16.42 Ma Tuff of Trout Creek Mountains, 16.33 Ma Tuff of Long Ridge, and 15.56 Ma Tuff of Whitehorse Creek[22, 24]. All eruptions resulted in caldera collapse, the largest being the 30 × 40 km McDermitt Caldera, the source for the ~1000 km³ Tuff of Long Ridge "supereruption" and location of the Kings Valley deposit (Fig. 2b).

To contextualize results from MVF, we also analyzed melt inclusions in rhyolites from volcanic centers in a variety of geologic settings around the world (Fig. 2a). We analyzed the 16.00 Ma peralkaline rhyolite Soldier Meadow Tuff from High

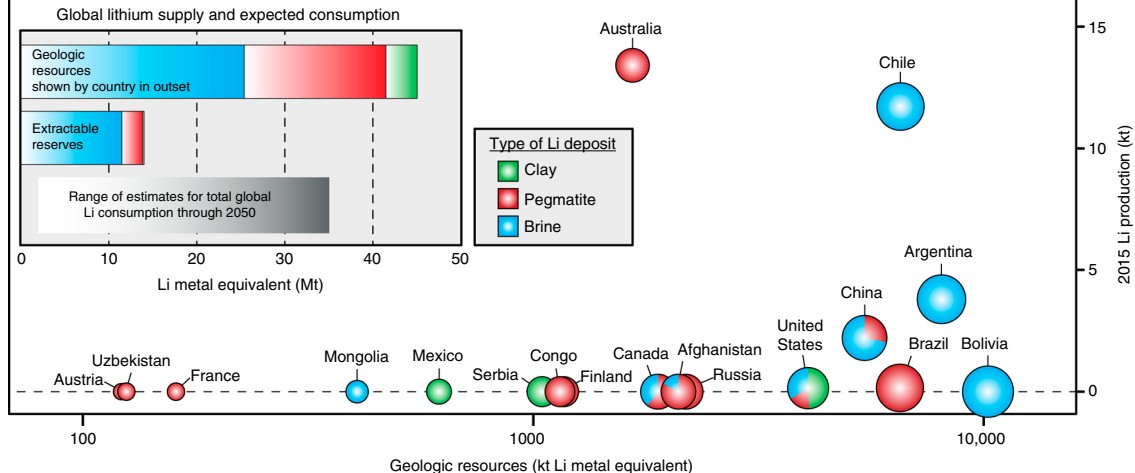

**Fig. 1** Global lithium resources and total 2015–2050 estimated consumption. Estimates for global Li resources are ~45 Mt Li and include brine, pegmatite, and clay resources[7, 10]. Global Li reserves (the amount of the resource that is currently economically extractable) are estimated at ~14 Mt Li, primarily from brine and clay deposits[7, 10]. Future Li consumption (2015–2050) estimates range anywhere from 3–35 Mt Li, depending on the efficiency of Li extraction and projections for electrification of the automobile industry[6, 8–10], highlighting the need to explore for new resources of Li

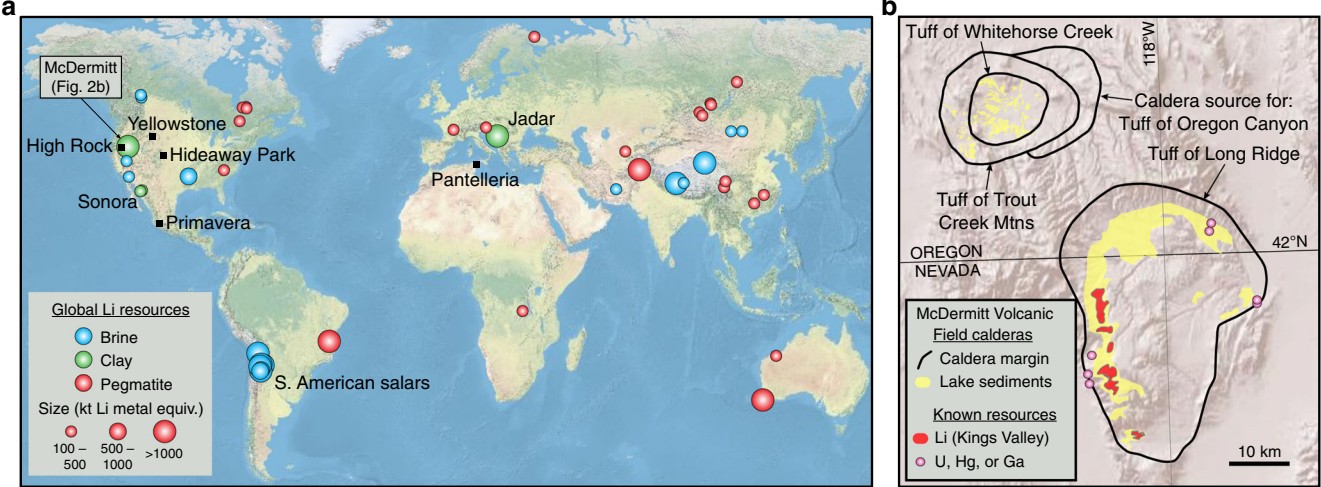

**Fig. 2** Location maps of global lithium resources and McDermitt Volcanic Field. **a** Map of worldwide Li brine, clay, and pegmatite resources larger than 100 kt Li[7, 10, 15] and locations of volcanic systems analyzed in this study (*black squares*). **b** McDermitt Volcanic Field calderas and associated caldera-forming ignimbrites[22] analyzed in this study. Also shown are outcrops of caldera lake sediments[15, 22] and locations of the Kings Valley Li deposit and Ga, U, and Hg resources in the McDermitt Caldera[15]

Rock caldera complex, which is another Middle Miocene center associated with Columbia river flood basalts, but lies southeast of MVF, where it formed on mafic island arc terranes accreted onto the craton margin[25]. We analyzed the weakly peralkaline 0.095 Ma Tala Tuff from the Pleistocene La Primavera caldera, Mexico, which also erupted through crust consisting of accreted terrains[26]. Thin, rifted continental crust is represented by a sample of a 0.014 Ma strongly peralkaline rhyolite lava from Pantelleria, a small volcanic island in the Strait of Sicily[27]. At the other end of the spectrum, we analyzed two samples from centers formed on thick, cratonic crust: the 2.14 Ma, alkali rhyolite Huckleberry Ridge Tuff from the Yellowstone volcanic field, Wyoming, and the 27.49 Ma peraluminous rhyolite Hideaway Park Tuff of the Red Mountain intrusive complex in Colorado[28].

**Measured Li concentrations**. Homogenized melt inclusions from rhyolite magmas of the MVF have average measured Li concentrations of ~1300 ppm, with the Tuff of Long Ridge having a slightly higher average Li concentration (~1500 ppm) than the Tuffs of Oregon Canyon (~1300 ppm), Whitehorse Creek (~1300 ppm), and Trout Creek Mountains (~1100 ppm) (Fig. 3; Table 1; Supplementary Data 2). Other magmas associated with the Yellowstone hotspot have similar average Li concentrations where erupted within felsic crust (~1200 ppm; Huckleberry Ridge Tuff) and lower average abundances where erupted through crust comprised entirely of accreted arc terranes at High Rock caldera complex (~400 ppm; Soldier Meadow Tuff) (Table 1). The peralkaline Tala Tuff, which like the Soldier Meadow Tuff, assimilated crust comprised of accreted island arc terranes, and has average measured Li content of ~400 ppm. Melt inclusions from a peralkaline rhyolite lava at Pantelleria have the lowest Li concentrations of all measured samples (~100 ppm on average), whereas the highest values (~5900 ppm on average) were measured on melt inclusions from the Hideaway Park Tuff.

The range of Li concentrations we obtained for Hideaway Park Tuff using the SHRIMP-RG (4200–8500 ppm) overlaps the range of concentrations determined by Mercer et al.[28] using LA-ICPMS on melt inclusions from the same sample (1000–6500 ppm)[11, 28]. We attribute the lower average values of Mercer et al.[28] to diffusive loss of Li during their extended homogenization experiments (330 min at ~900 °C). Using the estimated range of diffusion coefficients for Li along the c-axis in

quartz ($8.5 \times 10^{-8}$–$1.09 \times 10^{-6}$ cm$^2$/s)[29–33], we calculate that in 330 min at 900 °C, Li can diffuse ~60–210 microns. In contrast, similar calculations for the conditions under which we homogenized the same sample (25 min at ~1050 °C) yield diffusion distances of only ~15–60 microns.

Li concentrations in melt inclusions show positive correlations with incompatible elements (e.g., Rb; Fig. 3a) indicating that Li concentrations are highest in the most evolved melts. This is consistent with the strong incompatibility of Li in most minerals that crystallize in rhyolitic magmas, i.e., quartz, alkali feldspar, plagioclase, Na-amphibole, Fe-Ti oxides, zircon, and apatite. Only in biotite is Li weakly compatible ($D_{Li}^{biotite/melt} = 0.8$–$1.7$)[34, 35], but because biotite is absent or only a small proportion of the crystallizing assemblage, the bulk $K_D$ will always be much less than unity. This is supported by the observation that the only sample we studied with phenocrystic biotite (<1 vol.%)[28], the Hideaway Park Tuff, has the highest measured Li concentrations, which are positively correlated with the measured concentrations of the incompatible element Rb (Fig. 3a). Simple crystal fractionation models demonstrate that less than 50% fractional crystallization is required to explain observed ranges in Li and Rb concentrations in all the samples we studied (Fig. 3a).

**Correction for pre-entrapment vapor loss**. In order to compare magmatic melt concentrations of Li, we must adjust the measured concentrations for potential loss of Li to a magmatic vapor prior to melt inclusion entrapment, given the strong partitioning of Li into a vapor relative to coexisting melt ($D_{Li}^{vapor/melt} \approx 10$)[18]. We do this by examining the relationship between the melt inclusion Li concentrations vs. F/Cl ratios in the melt inclusions (Fig. 3b). In the absence of a coexisting vapor phase, Li concentrations will rise while F/Cl remains unchanged in the melt during differentiation of rhyolite magma, assuming all three elements behave incompatibly on crystallization. We believe this assumption is justified even though F and Cl are compatible in apatite and biotite[34, 36] because these minerals occur only as trace phases, <1 vol.% in crystallizing assemblages. As a result, their bulk $K_D$ values are less than unity and their crystallization will not have a discernable effect on the F/Cl of the residual melt. In the presence of a coexisting vapor, both Li and Cl partition into the vapor, whereas F largely remains in the melt, the result of a strong contrast in $D^{vapor/melt}$ for F (<0.4) and Cl (>10)[37].

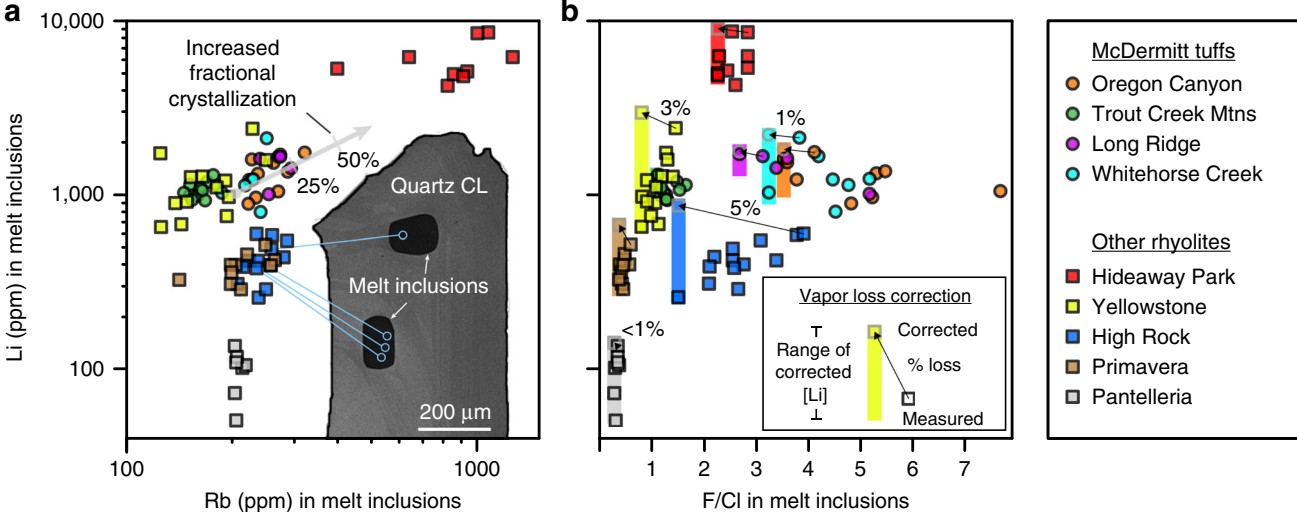

**Fig. 3** Measured and vapor-loss-corrected Li concentration in homogenized melt inclusions in quartz. **a** Li covaries with the strongly incompatible element Rb, consistent with it being incompatible in minerals crystallizing within rhyolite magmas, and indicating that it will be enriched by fractional crystallization. *Gray line* indicates 25 and 50% fractionation of a sample with 1000 ppm Li and 200 ppm Rb, assuming complete incompatibility. Also shown is a cathodoluminescent image (*CL*) of a quartz phenocryst with analyzed glass inclusions (*black* blebs) from the Soldier Meadow Tuff of the High Rock Caldera Complex, and locations of SHRIMP-RG pits for indicated data points. **b** Li vs. F/Cl measured in melt inclusions. Inclusions that have F/Cl greater than the smallest measured value for a given sample have lost vapor and Li prior to entrapment. The amount of Li and vapor loss for each inclusion, compared to the inclusion with the smallest F/Cl, was calculated (see Methods section, Supplementary Fig. 5) and the resulting range of corrected Li values for each sample are indicated by *vertical colored bars*. *Arrows* indicate examples of correction for individual analyses, with the associated calculated vapor loss by wt. % shown

**Table 1 Average Li concentrations in analyzed rhyolites**

| | Li (ppm) measured | Li$_{vc}$ (ppm) vapor-corrected |
|---|---|---|
| *McDermitt Volcanic Field (Nevada and Oregon)* | | |
| Tuff of Oregon Canyon | $1301 \pm 297$ | $1464 \pm 270$ |
| Tuff of Trout Creek Mountains | $1076 \pm 113$ | $1154 \pm 136$ |
| Tuff of Long Ridge | $1482 \pm 286$ | $1646 \pm 194$ |
| Tuff of Whitehorse Creek | $1315 \pm 438$ | $1479 \pm 467$ |
| average, McDermitt ignimbrites | $\mathbf{1294 \pm 167}$ | $\mathbf{1436 \pm 205}$ |
| | | |
| *Other rhyolitic centers* | | |
| Soldier Meadow Tuff, High Rock (Nevada) | $425 \pm 107$ | $553 \pm 192$ |
| Tala Tuff, Primavera (Mexico) | $389 \pm 70$ | $419 \pm 94$ |
| Huckleberry Ridge Tuff A, Yellowstone (Wyoming) | $1191 \pm 490$ | $1379 \pm 671$ |
| Hideaway Park Tuff (Colorado) | $5894 \pm 1577$ | $6319 \pm 1797$ |
| Pantelleria rhyolite lava (Italy) | $100 \pm 29$ | $106 \pm 33$ |

Data reported as average concentration of all melt inclusions analyzed from each sample ± one standard deviation. Bold values represent the average Li concentration and standard deviation for all ignimbrites from McDermitt Volcanic Field
Unit information listed in Supplementary Table 1
Full results listed in Supplementary Table 2

Any loss of Li to a vapor phase will therefore be accompanied by a rise in the F/Cl of the melt. If the amount of Li lost to a vapor is small relative to the predicted increase from fractional crystallization, the trend on a plot of Li vs. F/Cl will be positive, whereas if the loss of Li to a vapor outpaces fractionation, the melt inclusion data will form an array with a negative slope. To estimate the magnitude of vapor loss in each inclusion and initial magmatic Li concentrations prior to vapor loss (Li$_{vc}$), we employ the Rayleigh equation, assuming the lowest measured F/Cl in each sample is representative of the magma prior to degassing (see Methods section).

Measured inclusions from Pantelleria, Hideaway Park, and the Tuff of Trout Creek Mountains trapped melts that represent ~1% or less vapor loss by mass (Fig. 3b), leading to only moderate depletion in Li over the crystallization interval recorded by the melt inclusions (Table 1). The data arrays for melt inclusions from Primavera, Yellowstone, and High Rock are positive indicating that vapor loss (max. 5%; Fig. 3b; Supplementary Data 2) caused only a minor depletion in melt Li, which was more than compensated by the increase in melt concentration due to fractional crystallization (Fig. 3b). The negative data arrays for the Tuffs of Oregon Canyon, Long Ridge and Whitehorse Creek indicate that Li loss due to exsolution outpaced fractionation; nevertheless the relatively small range of F/Cl ratios recorded in each sample indicates that they lost no more than 4% vapor during the interval in which the melt inclusions were being trapped (Fig. 3b; Supplementary Data 2).

## Discussion

To determine the effect of tectonic setting on Li concentrations in rhyolites, we compare vapor-corrected Li concentrations (Li$_{vc}$) to Zr concentrations in melt inclusions, employing Zr as a proxy for the proportion of mantle-derived basalt relative to felsic crust in the make up of the rhyolite (Fig. 4a). We base this on the observation that Zr concentrations in rhyolitic magmas are controlled by the extent of zircon fractionation, either in a partially melted crustal source or during cooling and crystallization of magma. Saturation with respect to zircon is favored by lower melt temperature and greater melt polymerization[38]. Hence, magmas that contain a large fraction of low-temperature melts of felsic continental crust will have low concentrations of Zr due to their formation and subsequent differentiation in the presence of zircon. In contrast, magmas that form by melting mafic crust or evolve by fractionation of basaltic magma will tend to have higher concentrations of Zr, because zircon becomes stable only late in their evolution. By virtue of their bulk composition, strongly alkalic rhyolites cannot

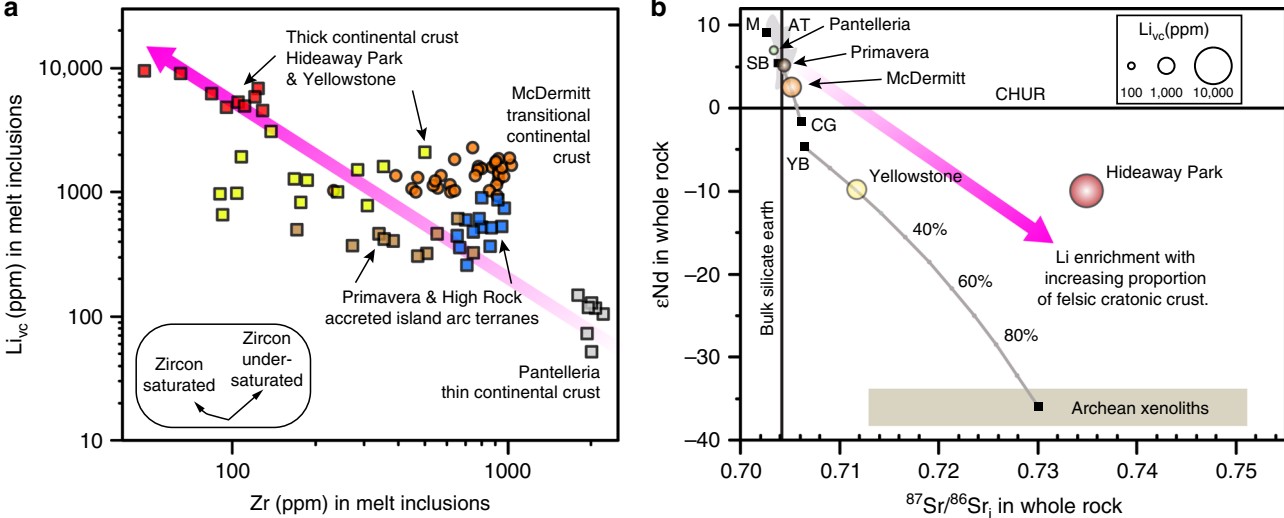

**Fig. 4** Li enrichment as a function of amount of felsic crustal material incorporated in the rhyolitic magma. **a** Li concentration (Li$_{vc}$, corrected for vapor loss) vs. Zr in melt inclusions. Li is highest in magmas in intracontinental settings that assimilate thick felsic crust. Inset box illustrates expected trends with increasing differentiation in the presence of zircon and in its absence. **b** Whole-rock εNd vs. $^{87}$Sr/$^{86}$Sr$_i$ of rhyolites with possible components (*M*: depleted mantle, *AT*: field for accreted island arc terranes, *SB*: Steens basalt, *YB*: Yellowstone basalt, *CG*: Cretaceous granite of the Santa Rosa Range, Nevada). The sizes of rhyolite data points correspond to average Li$_{vc}$ in melt inclusions. Simple isotopic mixing curves (*gray lines*) for Yellowstone and McDermitt demonstrate that these moderately Li-enriched magmas incorporated ~10% and ~50% felsic crust, respectively. Isotopic and concentration values and data sources appear in Supplementary Data 1 and 3

contain large proportions of partial melts of felsic crust, which are typically metaluminous to peraluminous in composition.

The lowest measured Li$_{vc}$ concentrations (~110 ppm) were found in the rhyolite from Pantelleria, which is strongly peralkaline and has the highest average Zr concentration (~2000 ppm) measured in the inclusions, reflecting formation of the rhyolite in thin, extended continental crust of the Strait of Sicily (Fig. 4a). On the other hand, the peraluminous Hideaway Park Tuff has the highest average Li$_{vc}$ concentrations (~6300 ppm) and lowest average Zr concentrations (~100 ppm), due to its derivation via partial melting of hybridized cratonic continental crust in an intracontinental setting[39]. Even a slight increase in the proportion of assimilated felsic cratonic material appears to be responsible for significant magmatic enrichment of Li. Magmas of MVF, which formed in crust transitional between the felsic Precambrian craton and accreted island arc terranes[22], have more than twice the amount of Li (Li$_{vc}$ ~1400 ppm) than similar weakly peralkaline (~700 ppm Zr) magmas at High Rock and Primavera that formed on crust comprised of mafic accreted arc terranes (Li$_{vc}$ ~600 ppm) (Fig. 4a).

These qualitative conclusions are supported by simple isotopic mixing models for Sr and Nd. The Pantelleria rhyolite lava, which contains the lowest abundances of Li$_{vc}$, has isotopic ratios very close to that of the depleted upper mantle (Fig. 4b), thereby implying negligible involvement of felsic continental crust in its formation. The low-Li Tala Tuff of Primavera, Mexico (300–600 ppm Li$_{vc}$) overlaps the isotopic field for accreted terranes (AT in Fig. 4b). The moderately Li-enriched magmas of McDermitt (~1400 ppm average Li$_{vc}$) are modeled as having incorporated ~50% transitional cratonic material ($^{87}$Sr/$^{86}$Sr = 0.70578, CG in Fig. 4b), similar to estimates made by modeling oxygen isotopic data[40]. The Yellowstone rhyolite magma is modeled as incorporating only ~20% Archean crust (Fig. 4b), consistent with previous isotopic estimates for the Huckleberry Ridge Tuff magma reservoir[41]. We conclude that the average Li$_{vc}$ concentrations are similar for Yellowstone and MVF (~1400 ppm), because the Archean crustal material incorporated

at Yellowstone is more felsic and richer in Li than the crust beneath MVF. This demonstrates that small proportions (~20%) of felsic continental crust or large proportions (~50%) of transitional continental crust yield similar Li enrichments in the resulting rhyolite magmas.

The similarity of the Nd isotopic ratios of magmas at Yellowstone and Hideaway Park suggests that the protolith from which the Hideaway Park partial melts were derived[42] had similar proportions of crustal and mantle material to Yellowstone. The high Sr isotopic ratio and peraluminous character of the Hideaway Park magma indicates that its crustal component must have been more highly radiogenic and aluminous, containing a higher proportion of micas and clays capable of having high concentrations of Li. Partial melts of this aluminous hybridized crust were therefore initially high in Li and other rare metals and were further enriched to levels measured in melt inclusions (4500–9400 ppm Li$_{vc}$) during 70–80% fractional crystallization[11, 39]. Similar concentrations have been measured on peraluminous non-homogenized (and thus more variable) melt inclusions from Spor Mountain, Utah (160–5200 ppm)[11], and for whole-rock peraluminous tin granites (4000–5200 ppm)[43]. Hideaway Park has higher Li concentrations than peraluminous obsidian from Macusani, Peru, which includes the highest Li concentration yet measured on degassed volcanic glass (maximum ~3400 ppm; average ~1300 ppm)[44, 45], and, along with other tin rhyolites in South America, flanks the world's largest Li resource (~10 Mt) at Salar de Uyuni, Bolivia (Figs. 1 and 2a)[7, 11].

Interestingly, average Li$_{vc}$ concentrations in peralkaline MVF melt inclusions (~1400 ppm) are higher than all Li concentrations ever measured on natural degassed volcanic glass aside from Macusani[46], and are comparable to the average value of glass from Macusani (~1300 ppm)[44, 47, 48]. They are also similar to concentrations found in homogenized melt inclusions in quartz from a tin-rich pegmatite (~1200 ppm)[43] and the average of five Mexican tin rhyolite lavas (~1400 ppm), though melt inclusions from individual lavas have average Li concentrations that range from ~500 to ~3300 ppm[19].

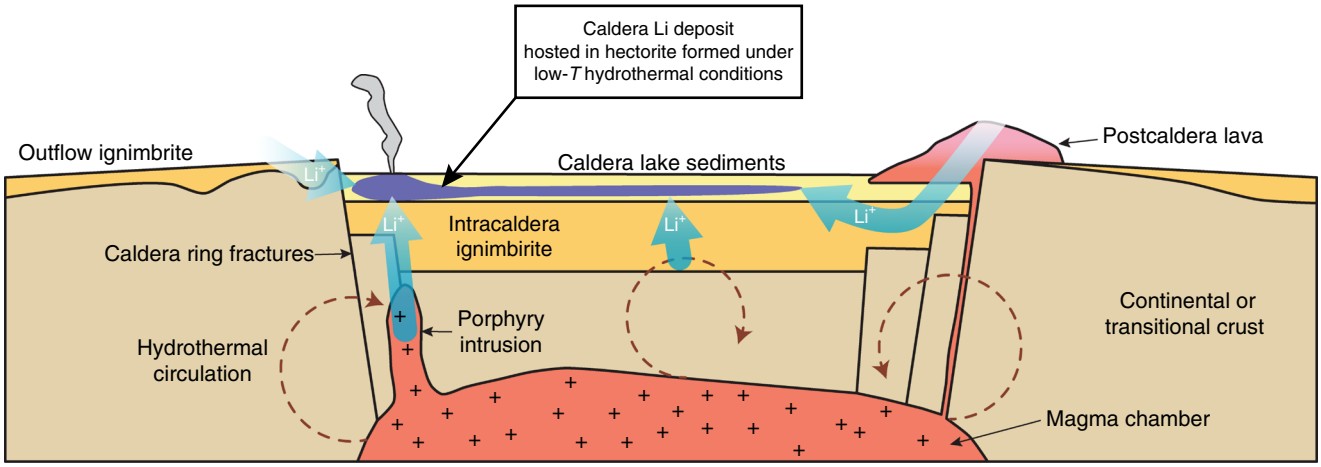

**Fig. 5** Schematic model for the formation of caldera-hosted Li clay deposits. Rhyolitic magmas in continental settings have elevated Li concentrations such that eruptions voluminous enough to result in caldera collapse produce volcanic products with sufficient total Li to form economic deposits. Post-caldera magmatism contributes additional Li via lavas and outgassing of intrusions; it also generates hydrothermal systems focused along caldera ring fractures. Li is leached from ignimbrite and caldera-related lavas by meteoric and hydrothermal fluids and is deposited in hectorite clays formed within ash-rich caldera lake sediments

Although the MVF magmas contain concentrations of Li (~1400 ppm $Li_{vc}$) that are 25–60% of values measured in some tin rhyolites and the Hideaway Park Tuff, the erupted volumes of the MVF magmas exceed those of the more Li-rich units by one to two orders of magnitude[19, 22, 28]. Hence, the total amount of Li available for a potential resource is significantly greater in large caldera settings than in the more Li-rich but smaller eruptive systems. We can make rough estimates of the amount of Li potentially available for leaching from intra- and extracaldera ignimbrite and subsequent deposit formation if we assume that (1) 45% of erupted Li is lost to the atmosphere during eruptive degassing, as observed in studies[11] that have compared Li concentrations of melt inclusions and matrix glass, and (2) following ignimbrite eruption only 20% of the outflow sheet is close enough to the caldera to be within its catchment watershed but 100% of the intracaldera ignimbrite is available for leaching. For the 1000 $km^3$ Tuff of Long Ridge, the eruption of which resulted in the formation of the McDermitt Caldera, this calculation yields an estimated maximum ~$2 \times 10^9$ Mt available for a caldera Li deposit. This is roughly an order of magnitude greater than our estimate for the maximum amount of Li erupted and available for leaching in the eruptive products at Hideaway Park ($3 \times 10^8$ Mt Li) and two orders of magnitude greater than available from a rhyolitic dome erupted in the tin belt of Mexico ($6 \times 10^6$ Mt Li).

In caldera settings (Fig. 5), in addition to leaching of the caldera-forming ignimbrite, a considerable amount of Li is supplied to the system by eruption of cogenetic post-caldera rhyolitic lavas and by degassing of Li-enriched magmatic fluids[20] exsolving from magma remaining in an underlying shallow magma chamber. Moreover, given that lakes and hydrothermal systems commonly form within intracontinental calderas, the caldera collapse feature itself serves as an ideal proximal basin for the accumulation of Li-enriched runoff and tuffaceous sediments, and the formation of a Li clay deposit on alteration of caldera lake sediments. This is illustrated in Fig. 5 as a proposed model for the Kings Valley Li deposit of the McDermitt Caldera. Eruption of the Tuff of Long Ridge at 16.33 Ma was accompanied by caldera collapse, and led to the formation of a caldera lake with active sedimentation. Intracaldera ignimbrite, nearby outflow sheets, and post-caldera explosive and effusive volcanism provided glassy rhyolitic material with moderate Li concentrations from which

meteoric water mobilized Li and deposited it within the closed basin of the caldera. Li-rich magmatic fluids exsolving from remaining magma rose through the western ring fracture zone for $10^5$ to $10^6$ years[22] and interacted with shallow meteoric water to form a near-neural, low-temperature hydrothermal system characterized by clinoptilolite-K feldspar alteration[13]. Li-bearing hectorite and illilte clays formed in this alteration zone, focused above caldera ring fractures where magmatic fluid influx was maximized (Figs. 2b and 5).

Although this model is based on observations from McDermitt caldera, most features are typical of rhyolitic calderas in intra-continental settings[49] and therefore can be applied more broadly. More than 100 other large Cenozoic calderas have been identified in western North America alone[50–52], including three within the northern part of the MVF (Fig. 2b)[22]. These calderas and others around the world that result from eruption of rhyolite formed by partial melting or assimilation of large amounts of transitional crust (e.g., McDermitt) or small amounts of cratonic crust (e.g., Yellowstone) are likely to contain large erupted volumes of moderately to extremely Li-enriched rhyolite. Of these centers, the largest calderas young enough to preserve hydrothermally altered caldera lake sediments have potential to host high-tonnage Li clay resources similar to the McDermitt/Kings Valley deposit. Exploration of these calderas could yield additional strategic Li resources for countries looking to reduce import reliance and help meet the rising global demand for lithium.

## Methods

**Homogenization of melt inclusions.** To minimize measurement artifacts due to post-entrapment processes and to obtain realistic compositions of entrapped melt, recent work has shown that homogenization of melt inclusions prior to in situ microanalysis is necessary[53, 54]. Post-entrapment cooling and decompression can cause separation of a magmatic vapor phase and/or crystallization of melt trapped within the host. As such, inclusions from most samples analyzed in this study were not initially homogenous glass. Commonly, inclusions had a bubble that comprised ~5–10% of the total volume of the melt inclusions (Supplementary Fig. 1). The presence of a vapor bubble is problematic for determining accurate melt concentrations of volatile phases (e.g., Li, $H_2O$, F, and Cl) and elements that readily partition into the vapor phase.

Given that Li is the focus of the present study and one of the most volatile alkali metals, we homogenized all melt inclusions. To determine the temperatures necessary to homogenize the inclusions, we used a heating stage mounted on a petrographic microscope at the United States Geological Survey in Menlo Park, California. Full homogenization of inclusions from four different samples at atmospheric pressure occurred between 750 and 950 °C, though some inclusions

never completely homogenized, presumably due to post-eruptive leaking. Using the upper homogenization temperature as a guide, initial batch homogenization experiments were performed at 1000 °C and 1 atm using a Deltech vertical-tube furnace at Stanford University. Under these ambient pressure conditions, approximately 75% of the inclusions cracked or leaked. To minimize inclusion failure, we homogenized samples using the ZHM (zirconium-hafnium-molybdenum) cold-seal pressure vessel operated by the United States Geological Survey in Menlo Park, California. For each experiment, an ~18 mm Au capsule was filled with ~0.1 g of quartz phenocrysts containing melt inclusions and crimped (not sealed) shut at the top. Capsules were individually loaded into the ZHM vessel and pumped to a pressure of 1000 bar using Ar gas as the pressure medium. Once this pressure was reached and stability was ensured, the pressure vessel was lowered into a Deltech DT31VT resistance furnace calibrated following established methods[55]. A Pt-Pt$_{90}$-Rh$_{10}$ thermocouple was used to monitor the temperature every 5 min during the experiment. Samples reached the target homogenization temperature of 1000 °C after approximately 30 min and were kept at that temperature for only ~25 min to minimize diffusive loss of small monovalent cations from the inclusions (e.g., Li$^+$, H$^+$, Na$^+$, and Cu$^+$)[32]. At this time, the pressure vessel was removed from the furnace and immediately inverted so that the Au capsule fell to the water-cooled head of the vessel, quenching the melt inclusions to homogenous glass. This homogenization procedure is nearly identical to that of refs. [11, 28]. Post-experiment imaging of inclusions show that vapor bubbles and crystals disappeared in >90% of inclusions (Supplementary Fig. 1) and <10% of quartz phenocrysts were cracked.

Quartz phenocrysts hosting the homogenized melt inclusions were mounted in crystal bond and polished (using a diamond solution) until inclusion(s) within individual crystals were exposed. Polished phenocrysts were then removed from the crystal bond and remounted in epoxy alongside RLS 37, 132, 140, 158, Macusani[44], NIST SRM 613 and 615, and ATHO-G[56] glass standards. Macusani was included as a reference material with high Li content (3400 ppm). The polished mount was gold coated and imaged with a JEOL 5600 scanning electron microscope at Stanford University (Supplementary Fig. 1).

### Confirmation of homogenization technique.
To demonstrate that melt inclusion analysis is necessary for measuring accurate magmatic concentrations of Li, we analyzed matrix glass and homogenized melt inclusions from a peralkaline rhyolite lava at Pantelleria, Italy[27]. Concentrations of Li, S, and Cu are significantly lower (~60% each) in the matrix glass relative to the melt inclusions, suggesting that post-entrapment degassing of the magma and/or lava depleted the melt in these volatile elements (Supplementary Fig. 3). All other analyzed elements are enriched in the matrix glass relative to the melt inclusions, indicative of post-entrapment melt evolution (Supplementary Fig. 3). The observed range in matrix/inclusion ratios among the elements that were not volatilized is explained by relative incompatibility of the elements during post-entrapment evolution of the melt; elements closer to unity (e.g., Zr) behave more compatibly than elements with higher matrix/inclusion values (e.g, Ti and Rb). These results show that analyzing melt inclusions is necessary for any study on the original magmatic Li (or S and Cu) concentrations of magmas. In the case of this rhyolite lava at Pantelleria, 42% of the Li in the melt was volatilized and segregated from the melt after inclusion entrapment. These observations agree with the results of Hofstra et al.[11] who calculated inclusion-matrix glass Li depletions ranging from ~36–53% (average 45%) in samples of ignimbrite from the Valles Caldera, New Mexico[21].

To demonstrate the necessity of homogenizing melt inclusions for obtaining accurate Li concentrations, we analyzed non-homogenized and homogenized inclusions (Supplementary Fig. 1) from the peralkaline Soldier Meadow Tuff of the Mid-Miocene High Rock caldera complex[25]. Results confirm that non-homogenized samples have much lower concentrations of small monovalent cations Cu$^+$ and Li$^+$ (Supplementary Fig. 4) due to the partitioning of Li into vapor bubbles. The presence of vapor and/or crystals in non-homogenized inclusions likely also leads to slightly higher concentrations of vapor and crystal incompatible elements in analyzed glass; these phases are not present in the glassy homogenized melt inclusions. Homogenization of melt inclusions is therefore necessary for accurately quantifying the concentration of Li in magmas.

To test the effectiveness of our homogenization experiments, we analyzed large inclusions in multiple locations (Fig. 3a). Results demonstrate that the vast majority of these inclusions are homogenous within analytical uncertainty (Supplementary Data 2). We therefore are confident that even with only ~25 min at 1000 °C, our homogenization procedure is effective at homogenizing melt inclusions.

### SHRIMP-RG analysis.
Standards and unknowns were analyzed on the Stanford-U. S. Geological Survey sensitive high-resolution ion microprobe with reverse geometry (SHRIMP-RG) at Stanford University in two separate sessions during April 2014 and March 2015. Secondary ions, accelerated at 10 kV, were sputtered from the target spot using an O$^{2-}$ primary ion beam with an intensity varying from 0.7 to 1.0 nA. The primary ion beam spot had a diameter between 12–16 microns and a depth of ~1 microns. The acquisition routine included analysis of $^7$Li$^+$, $^9$Be$^+$, $^{11}$B$^+$, $^{19}$F$^+$, $^{30}$Si$^+$, $^{32}$S$^+$, $^{35}$Cl$^+$, $^{29}$Si$^{16}$O$^+$, $^{49}$Ti$^+$, $^{54}$Fe$^+$, $^{63}$Cu$^+$, $^{69}$Ga$^+$, $^{85}$Rb$^+$, $^{88}$Sr$^+$, $^{89}$Y$^+$, $^{90}$Zr$^+$, $^{93}$Nb$^+$, $^{138}$Ba$^+$, $^{139}$La$^+$, $^{140}$Ce$^+$, $^{146}$Nd$^+$, $^{147}$Sm$^+$, $^{151}$Eu$^+$, $^{158}$Gd$^{16}$O$^+$, $^{159}$Tb$^{16}$O$^+$, $^{162}$Dy$^{16}$O$^+$, $^{166}$Er$^{16}$O$^+$, $^{172}$Yb$^{16}$O$^+$, $^{208}$Pb$^+$, $^{232}$Th$^{16}$O$^+$, and $^{238}$U$^{16}$O$^+$. Analyses were performed using a single scan by peak-hopping through

the mass table, and each mass is measured on a single EPT$^®$ discrete-dynode electron multiplier operated in pulse counting mode. Count times for trace-element measurements ranged from 2 to 12 s to optimize counting statistics for each isotope. The background for the electron multiplier is very low (<0.05 cps), and is statistically insignificant for the trace elements reported in this study.

Measurements were performed at mass resolutions of $M/\Delta M = $ ~9800 (10% peak height measured on $^{85}$Rb) to resolve interfering molecular species from the masses of interest, particularly for REE. Heavy rare earth elements (HREE) are measured as oxides because metal ions can contain isobaric interferences that often cannot be fully resolved, and which are not present for the oxides at higher mass. To further minimize the intensity of molecular interferences, the SHRIMP-RG was operated using the energy selection window to only accept high-energy ions into the collector (~40 V offset). Because metal ions (e.g., Pb$^+$) have higher energy than molecules with the same mass, this procedure dramatically reduced potential isobaric interferences.

Count rates of each element were ratioed to $^{29}$Si$^{16}$O to account for any primary current drift, and derived ratios for the unknowns are compared to an average of those for the standards to determine concentrations. Calibration curves for the 2014 and 2015 SHRIMP-RG sessions were plotted using measured ratios and published concentrations[44, 56] for standard glasses (Supplementary Fig. 2). For each element, calibration curves were calculated using the best-fit line of the average and standard deviation of all values measured for each standard glass (error bars are often smaller than the width of the data point in Supplementary Fig. 2). Data from synthetic NIST SRM glasses (611, 613, and 615) were only included if there was insufficient published values for natural glass standards to produce a calibration curve. We considered the natural glasses to be preferable because they are similar in composition to unknowns, and the NIST SRM synthetic glasses often define a slightly different calibration trend from natural sample calibrations, which we attribute to matrix effects. Supplementary Fig. 2 shows calibration curves for elements used in the main text of the manuscript (Li, F, Cl, Rb, and Zr). Analytical errors are derived from the 68% confidence bands about the linear fit of the calibration curve (shown in Supplementary Fig. 2 as light blue for the Fall 2014 calibration and light pink for the Spring 2015 calibration), which we feel is appropriate because it reflects the reproducibility of the standard materials. Concentration and errors for all elements are listed in Supplementary Data 2.

Based on post-analysis inspection of the melt inclusions using the JEOL 5600 scanning electron microscope, individual analyses were considered compromised and excluded based on the following criteria: (1) cracks emanating in host from inclusions: likely lost volatile elements during eruption or homogenization experiments; (2) inclusions on edge of host quartz: glass located on the edges of the crystals could be re-entrants, and therefore not representative of pre-eruptive magma; (3) small inclusions: the composition of inclusions less than 20 microns in diameter are likely strongly affected by boundary-layer effects[20, 57, 58]; (4) non-homogenized inclusions: inclusions with a vapor bubble or crystals still visible due to incomplete homogenization; (5) primary beam overlap with host: inclusions where the analytical pit overlapped the quartz host resulting in depletion of both incompatible and compatible elements. This systematic inspection resulted in the exclusion of 66 of 150 analyses. All melt inclusion glasses analyzed in this study are vapor bubble- and crystal-free.

### Vapor loss correction.
To estimate the fraction of vapor loss in a given sample and initial Li concentrations prior to vapor loss (Li$_{vc}$), we employ the Rayleigh equation for Li, F, and Cl:

$$E_{vc} = \frac{E_m}{f^{D_E - 1}}, \qquad (1)$$

where $E_{vc}$ = concentration of element E (Li, F, or Cl) prior to vapor loss, $E_m$ = measured concentration of element E, $D_E$ = vapor/melt partition coefficient of element E, and $f$ = fraction of melt (1−$f$ is fraction of vapor). Assuming $D_{Li} = 10$, $D_F = 0.1$, and $D_{Cl} = 20$,[18, 37] we combine Rayleigh equations for all three elements to establish:

$$Li_{vc} = \left[ \frac{F_m * Li_m^{2.22}}{Cl_m * X} \right]^{\frac{1}{2.22}}, \qquad (2)$$

where $X = F_{vc}/Cl_{vc}$. Assuming that the lowest measured value of $X$ for a given sample is representative of the magma prior to degassing, we calculate Li$_{vc}$ for each inclusion as being a "vapor-corrected" Li concentration prior to vapor loss. With values of Li$_{vc}$ and Li$_m$ known for each inclusion, we can then calculate the fraction of melt present ($f$) and vapor lost (1−$f$) for each inclusion using a rearrangement of the Rayleigh equation:

$$f = \left( \frac{Li_m}{Li_{vc}} \right)^{\frac{1}{9}}. \qquad (3)$$

A schematic representation of these equations is shown in Supplementary Fig. 5. Data that plot vertically from the inclusion with the lowest F$_m$/Cl$_m$ have no variation in F/Cl and therefore have experienced no degassing, just progressive evolution indicated by an increase in incompatible element Li. If data with F$_m$/Cl$_m$

values increasing from the lowest measured $F_m/Cl_m$ value fall along a single exponential curve (the slope of which is determined by the partition coefficient of Li, here depicted as $D_{Li} = 10$ in solid black lines with example slopes of $D_{Li} = 1$ and $D_{Li} = 20$ shown as gray dashed lines), they have undergone pure degassing from a melt of constant composition (e.g., Pantelleria, Fig. 2a). Data that fall off these two end members either show negative slopes, in which case they are dominated by degassing with only minimal evolution, or positive slopes which indicate increasing amounts of degassing as evolution progresses.

We stress that the application of this methodology to the samples relies on several assumptions. First; we assume that no vapor was lost in the inclusion from each sample with the lowest $F_m/Cl_m$. For a given sample, if the inclusion with the lowest $F_m/Cl_m$ was trapped after vapor exsolution already began, the calculated percent vapor loss and $Li_{vc}$ for each inclusion represents minimum values. Second; we assume that the partitioning behavior of Li, Cl, and F remains constant throughout the whole time quartz is crystallizing and trapping inclusions. For all elements and samples, concentrations of Li, Cl, and F increase at roughly constant slopes with increasing concentrations of incompatible element Rb (and Zr for peralkaline samples). This indicates that all three elements retain approximately the same partitioning behavior throughout the recorded quartz crystallization history of all analyzed magmas. Finally, we assume that the bulk partition coefficients for Li, Cl, and F remain below unity for all analyzed magmas. Though Cl and F are compatible in apatite and biotite ± hornblende[34, 36, 59], and Li is weakly compatible in biotite[34], these phenocrysts exclusively occur as trace phases (less than 1 vol.%) in rhyolitic magmas. For phenocryst assemblages of all magmas analyzed in this study, we calculate that bulk partition coefficients ($K_D$) for all three elements are significantly less than unity, consistent with incompatible behavior. We are therefore confident that the observed ranges in F/Cl are recording vapor loss and not crystallization processes.

**Data Availability**. The authors declare that all data generated or analyzed in this study are included in the published article and supplementary data files. Supplementary Data 1–3 contain all data discussed in article and used to generate figures in the main text (Figs. 1–5) and in the supplementary figures (Supplementary Figs. 1–5).

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

## Acknowledgements

The authors thank J. Lowenstern and T. Sisson (US Geological Survey, Menlo Park, CA), and J. Stebbins (Stanford University) for offering their labs and assistance with homogenization experiments, W. Hildreth (US Geological Survey, Menlo Park, CA) for providing the sample of Huckleberry Ridge Tuff, A. Hofstra and C. Mercer (US Geological Survey, Lakewood, CO) for providing the sample of Hideaway Park Tuff, and M. Grove and E. Miller (Stanford University) for fruitful discussions. T.R.B. was partially supported by a US Department of Defense NDSEG Fellowship.

## Author contributions

T.R.B. designed the study with assistance from M.A.C., J.J.R. and G.A.M. T.R.B. collected samples of the Tuffs of Oregon Canyon, Trout Creek Mountains, and Whitehorse Creek, G.A.M. provided samples from Pantelleria and Primavera, J.J.R. provided the Tuff of Long Ridge sample, and M.A.C. provided the Soldier Meadow Tuff sample. T.R.B. prepared and analyzed all samples with assistance from M.A.C. T.R.B. took the lead in interpreting results and writing the paper, with substantial contributions by G.A.M. and M.A.C. and helpful comments by J.J.R.

## Additional information

**Competing interests:** The authors declare no competing financial interests.

