## [Peer Review File · Nature Communications]

Reviewers' Comments:

Reviewer #1 (Remarks to the Author)

The manuscript presents a collection of data intended to underscore the significance of Li in clay deposits from felsic volcanic calderas. The significance of these clay-rich deposits has been recognized but perhaps not fully appreciated. Although Li as a resource stands to be in high demand, the most recent USGS prospectus suggests that with currently identified deposits society is adequately supplied for the coming century; this tends to undercut the implicit point of the manuscript that there is a critical need for identifying other deposits.

From a scientific standpoint, the manuscript is quite sound. They use an interesting approach to address the question of how much Li may have originally been present in a given felsic magmatic system (given potential loss in exsolved vapor). The general relationship between, for example, Zr and Li is very clever and appears suited for use as an exploration tool of sorts. The authors should be commended for producing excellent graphics.

In the end I think the work is solid but lacks that hook that is supposed to be there for Nature. I think the work, in a fuller form, would be an outstanding contribution to a more discipline-focused journal like *Econ Geol* or *JVGR* or even *Am Min*.

In include some observations related to lines in the text below.

line 39: What is a "water-rich mineral"? Spodumene is anhydrous, so I don't see how this statement makes sense.

line 66: It is general practice to avoid beginning sentences with abbreviations (so the element name would be spelled out here).

line 70: Is Li "strongly incompatible" in a Na-amphibole? I would think its incompatibility would be somewhat more limited.

line 74: One of the reasons you stated above for dealing with melt incs is because of pre-eruptive degassing. If that was so, then why would you need to correct?

line 118: What is "it" here?

line 124: Also Sr, Nd isotopic or others?

line 126: You mean enriched in Rb relative to Sr, or having a high Rb/Sr, right?

line 134: Perhaps a more primary reference for tin granites, which are rather variable in composition, would be better here.

line 136: This concentration is suspiciously like that reported earlier by London et al. (1989).

Reviewer #2 (Remarks to the Author)

January 7, 2017

Review of NCOMMS-16-29348

Adam Simon

This paper presents and interprets new data for the enrichment of Li in rhyolitic magmas, and stresses the importance of such rocks as a future resource of Li. As Li is considered a critical metal necessary for production of Li batteries, and worldwide consumption of Li is expected to increase in the next decades, the paper is appropriate for a wide audience. It is not stressed much in the paper, but it is incredibly important to diversify the global Li supply chain, which at present is dominated by a very small number of countries. The situation is not as dire as it is for rare earth

metals (Chinese monopoly), but most of the EU and the United States are almost entirely reliant on Li imports. For example, the United States imports almost two-thirds of consumed Li. For most countries in the EU, the import reliance is higher. Thus, identifying 'new' resources of Li that are likely economically viable from a mining perspective is important. Overall, the paper is easy to read and the interpretations are rooted in the data presented. My comments below are based on reading the paper in sequence. If I make a comment that is subsequently addressed in the paper, please forgive that.

L26: The concept of "resource" is likely unknown to readers of Nature Communications. I suggest to define this somehow in the space allowed. Ideally, it would be great to present the economic reserves for Li for each type of resource to give the reader a sense of how price plays a role in making any particular resource economical to extract.

This is a huge range for Li resources. I checked the USGS December 2016 annual report for Li resources and they quote 34 million tonnes total worldwide. Where does the upper estimate of 65 million tonnes originate?

<https://minerals.usgs.gov/minerals/pubs/commodity/lithium/mcs-2016-lithi.pdf>

Perhaps something else would be to report the reserves / production for each of the resources to give the reader perspective on the expected depletion rate of a particular resource. Right now brines are cheap, but they'll be exhausted in the near future as the Gigafactory and others around the world produce more Li batteries. Also, Li sources other than brines are becoming economically competitive. I know the authors are space limited, but it would be great to add a sentence or few (or a table) that shows the reader the expected lifetime of each Li resources relative to forecasted consumption. Those are available via the USGS annual reports and other free sources. Most people simply assume an inexhaustible source of metals, but there are limits. And as one source becomes exhausted, others become economical.

This is the main selling point to a wider audience who (like it or not) do not care about how or why the rhyolitic magmas are enriched in Li. The mining community cares if it helps exploration strategies. The USGS certainly cares as they are charged (among other things) to identify reserves of strategic/critical metals. The business community (Tesla) cares only that sustainable Li resources exist to build their battery. So, some of my comments (including the one above) are meant to help the paper out of the petrology silo.

L40-41: In 2015, this was not true. The spodumene Li mining operations in Australia gained market share and were the second largest producer of Li to global markets. In fact, more Li was mined from spodumene than was mined from the salars in Chile (note that production from Bolivia continues to have zero supply to global markets). You should revise your statement here as market forces are reshaping the economics of the Li supply chain.

L43-44: The writing is confusing. Granite and rhyolite are essentially compositionally identical. What is different about a rhyolitic magma chamber and a granitic magma chamber? As written, a separate magmatic volatile phase percolates and accumulates in the top of a rhyolitic magma chamber, but not a granitic chamber. And likewise, bubbles only accumulate at the top of a rhyolitic magma chamber, but not a granitic magma chamber. Is this true? I have seen lots of granitic plutons that experienced volatile exsolution and loss of a magmatic volatile phase to the surrounding rock.

L62: Does "rhyolitic magma" equal "rhyolitic whole rock", where you assume that the whole rock represents the magma that existed prior to eruption? That is, there was no mass loss to a vapor phase prior to and/or during eruption? Perhaps it is appropriate to use "whole rock" as that is what you measured.

L75: Is the partition coefficient for Li static, or a function of melt composition, which, in turn, at a given pressure and temperature controls the composition of the vapor phase? In this case, the total Cl concentration of the vapor phase, which is important for scavenging Li from the melt. Do you have any independent (i.e., fluid inclusion) evidence for the salinity of the vapor phase in your samples? Are there any co-entrapped fluid and melt inclusions in the quartz phenocrysts?

L77: Unless apatite is part of the crystallizing assemblage, whereupon the F/Cl of the melt will decrease with fractionation. So this assumes apatite is not a stable phase. Was apatite absent during fractionation of all the systems?

L94: Was the oxidation state of all the systems similar? This is important when comparing potential loss of sulfur to a volatile phase, a process that is strongly redox dependent. A slight shift of oxidation state across the sulfide – sulfate transition will decrease the vapor/melt partition coefficient by at least one order of magnitude.

Apatite can also incorporate sulfur, and this is highly dependent on the oxidation state of the system (see new paper by Konecke et al., 2017, *American Mineralogist*: Co-variability of S⁶⁺, S⁴⁺, and S²⁻ in apatite as a function of oxidation state: Implications for a new oxybarometer).

L 101: Should be singular tense as a magma cannot have more than one concentration of Zr (unless you refer to the concentration of Li in discrete phases of the magma).

L131-133: I'm not sure I entirely follow the first part of the sentence. Do you mean that you measured Li in the Hideaway Park whole rock by solution chemistry and also by fusion LA-ICPMS of a glass bead? Why the half order of magnitude range of Li concentrations? And one order of magnitude variability for Spor Mountain samples?

L135: As written, some analyses of Hideaway Park have higher Li concentrations than Macusani, but other samples have lower concentrations. So this statement is not strictly true. What is the average and standard deviation of the Hideaway Park samples? Perhaps the concentrations of the two locations are statistically similar?

L136: Considering that these are tin granites, which are known to evolve at reduced fO₂, relative to typical arc magma systems, is there anything about the oxidation state of the systems that may have played a role in controlling the fractionating assemblage that, in turn, led to Li enrichment?

L146: But this depends on the total tonnage of the resource host rocks, and not simply the concentration of Li.

L154: Lavas of any composition? Or evolved, rhyolitic lavas?

L155: "...a subjacent shallow magma chamber". There is a lot of discussion among volcanologists and folks in economic geology as to whether the addition of such volatiles from deeper levels of the volcanic plumbing system are in fact required to drive eruption of the magma that results in caldera formation.

L170: Is this accurate? As written, a rhyolitic magma physically assimilates a lithospheric melt. Do you perhaps mean that the rhyolitic magma itself is the product of lithospheric melting? Revise to readability.

Supplement

I know and respect Paul Wallace, but he is far from the first person to report that rehomogenization of melt inclusions is necessary. Folks in economic geology spent years convincing volcanologists that the vapor bubble is not a vacuum. The work of Rosario Esposito and

Bob Bodnar is perhaps the most convincing over the past few years (Esposito et al., 2014. An assessment of the reliability of melt inclusions as recorders of the pre-eruptive volatile content of magmas, *Am Min*). It is also possible to ablate the entire inclusion without rehomogenization and obtain accurate compositions of the inclusion. Especially when quartz is the host and Li is incompatible in quartz, this should be straightforward. Only if you are analyzing a highly volatile fugitive component of the inclusion (e.g., CO₂, CH₄, S) is it critical to homogenize. But I am used to doing this with a larger diameter laser ablation ICP-MS where it is easy to ablate the inclusion and a small volume of host material and deconvolve the mixed signal. So it is not relevant for your analytical technique.

Did you do any analytical traverses away from a rehomogenized inclusion away into the quartz host to assess diffusive loss of Li or other elements either prior to sampling or during rehomogenization?

Did you work on melt inclusion assemblages? How did you assess that the inclusions were trapped at the same stage of quartz growth?

In Figure S3, what are the 3 inclusions that you annotate with "trapped other magma"? Are those inclusions spatially associated with other inclusions for which data are presented? Do you imply that the quartz host for some inclusions was recycled during magma mixing from another magma?

When you refer to post-entrapment degassing and loss of Li, S, Cu, technically, it is the melt phase that reaches volatile saturation and degasses.

Figure S4. Another semantics issue, but I would not say "elements volatilized on eruption". The melt reaches volatile saturation and those volatile-compatible elements are partitioned from the melt to the volatile phase, which is the driver for the eruption (i.e., the mass transfer of volatiles from melt to fluid occurs prior to eruption). Yes, some additional loss occurs during eruptive decompression, but much occurs prior to eruption.

Are the partition coefficient values you assume appropriate for your bulk compositions? The partitioning of Cl is highly sensitive to changing melt composition, temperature and pressure. Also, the acidity of the exsolved vapor is strongly dependent on the alkalinity of the silicate melt and, in turn, can affect Li partitioning.

Is apatite never present in any of the systems? Apatite is seemingly ubiquitous in intermediate to felsic magmas. Further, the partitioning of F and Cl between silicate melt and apatite depends on melt composition, pressure and temperature. I am not intimately familiar with the rocks in this study, but find it surprising that apatite is not present.

Reviewer #3 (Remarks to the Author)

This interesting manuscript provides important new data on lithium in volcanic systems that better support modeling of the generation of lacustrine-based Li deposits located in volcanic calderas. The melt inclusion data are of high quality and are convincingly applied to Li deposit formation.

This manuscript is worthy of publication after minor revision as detailed below.

Major comments:

I. The abstract could be rewritten to better relate the primary data source of this study (i.e., melt inclusions) to the larger processes of Li concentration in sedimentary environments. For example, it does not logically follow (for non-geological background readers) that measurements of Li in silicate melt inclusions provide direct constraints on Li concentrations/deposits in intracontinental settings, see lines 15-18, given that other processes (weathering of volcanic materials and the precipitation of Li-enriched clay minerals) are involved. Folks with geo-backgrounds will be able to understand this, but given the importance of Li-bearing batteries (and current interest in related energy storage issues), presumably other readers will be interested in this paper when published so it will be useful to expend extra effort to make the abstract more broadly understandable in this context.

II. In the Supplementary Information 1 document, the authors apply (F/Cl) ratios to constrain the likelihood and quantity of magmatic vapors in the systems investigated. The slopes of (F/Cl) plots are interpreted in this regard. This approach assumes that fractional crystallization (or other potential processes such as magma mixing) has no measurable influence on the behaviors of F, Cl, and Li at the stage of vapor saturation (specifically, that Li, Cl, and F increase similarly with fractional crystallization and/or that F is not fractionated relative to Cl by any process other than loss to fluid). This does not, necessarily, seem to be a reasonable assumption for rhyolitic systems, and should be better justified/explained in the revised manuscript.

Also, to this point, it would be useful for the authors to include any previously published constraints on the likelihood and timing (and quantities, if estimated) of a vapor phase in these magmatic systems studied, if the vapor phase was previously proposed/determined to exist - for the revised manuscript - given the importance of a magmatic vapor phase in Li transport.

Minor, annotated comments in the text pdf, by line numbers:

1. 37-39: spodumene is not a water-rich mineral; text should be corrected.
2. 41-44: given the prior statement about Li concentrating in lepidolite and spodumene, it is important to distinguish crystallization of the dominant (non-rare) silicate minerals such as feldspar, quartz, non-Li bearing micas, etc. (for the non-geology reader) - otherwise this might appear contradictory. For example, this is better explained in lines 69-70.
3. 62-66: again for the non-geo reader, please specify explicitly if the Li contents of rhyolitic magmas are melt inclusion data (not whole-rock data). Also, melt inclusions provide constraints on melt compositions and not constraints for phenocryst-bearing magma compositions, to be specific, since the magma represents melt and crystals. Similarly, in the supplementary table that describes the various volcanic fields (and rock sources), please specify that the average Li contents determined in this study are from melt inclusion analyses.
4. 143: sample 90-5.1.1 of Webster et al. (1996) contains 0.33 wt% Li on average (see Table 4).
5. Other: In the Supplementary Information 1 document, the results of rehomogenization of the melt inclusions are described texturally, but please explicitly state whether, or not, only bubble-free and crystal-free melt inclusions were analyzed for this study.
6. Other: In the supplementary table that describes the various volcanic fields (and rock sources), it would be very useful/helpful if some basic compositional information was included for row 4 (sample types)... brief information on alkalinity/aluminosity should be added.
7. The figure axis labels for Figures 2 and 3 should explicitly state "Li (ppm) in melt inclusions" and "Rb (ppm) in melt inclusions", "Nd epsilon in whole rocks", etc.
8. Does Figure 4 represent a generic/general model as the figure implies or does it specifically represent the Kings Valley Li deposit of the McDermitt caldera (as text lines 158-159 state)? I ask this because the diagram includes a porphyritic intrusion and I am curious to know if this intrusion is broadly applicable to caldera Li deposits in general or it just applies to the Kings Valley deposit/caldera?

Jim Webster

Reviewer #1 (Remarks to the Author):

The manuscript presents a collection of data intended to underscore the significance of Li in clay deposits from felsic volcanic calderas. The significance of these clay-rich deposits has been recognized but perhaps not fully appreciated. Although Li as a resource stands to be in high demand, the most recent USGS prospectus suggests that with currently identified deposits society is adequately supplied for the coming century; this tends to undercut the implicit point of the manuscript that there is a critical need for identifying other deposits.

We disagree with the conclusion of Reviewer #1 that Li is adequately supplied for the next century. This is only true if Li consumption remains constant at current levels. Most recent assessments (Kesler et al., 2012; Vikström et al., 2013; Pehlken et al., 2015) have the demand for Li increasing substantially over the next century, and that by 2050 the demand will account for 20% to 250% of the current global reserve. We reworded the abstract (**lines 11 – 22**) and introduction (**lines 29 – 36**) to state more explicitly the economic necessity of identifying more Li deposits. In addition, as suggested by Reviewer #2, we include new wording on the economic importance of diversifying the global Li supply chain from the current over-reliance on production from two countries (**lines 34 – 36**). We added a new Figure 1 to illustrate these points.

From a scientific standpoint, the manuscript is quite sound. They use an interesting approach to address the question of how much Li may have originally been present in a given felsic magmatic system (given potential loss in exsolved vapor). The general relationship between, for example, Zr and Li is very clever and appears suited for use as an exploration tool of sorts. The authors should be commended for producing excellent graphics.

In the end I think the work is solid but lacks that hook that is supposed to be there for Nature. I think the work, in a fuller form, would be an outstanding contribution to a more discipline-focused journal like Econ Geol or JVGR or even Am Min.

As noted above, we extensively revised the text of the abstract and introduction to emphasize the motivation for this work and the importance of the results. As such, we believe the paper has broad appeal, including those working in mineral economics, mineral exploration, and strategic mineral policy, as well as researchers in ore deposits, volcanology, petrology, and mineralogy.

In include some observations related to lines in the text below.

line 39: What is a “water-rich mineral”? Spodumene is anhydrous, so I don’t see how this statement makes sense.

We corrected this sentence by eliminating “water-rich”.

line 66: It is general practice to avoid beginning sentences with abbreviations (so the element name would be spelled out here).

We now spell out the element name (**line 37**).

line 70: Is Li “strongly incompatible” in a Na-amphibole? I would think its incompatibility would be somewhat more limited.

Lithium is slightly more compatible in amphibole than in feldspar, quartz, and Fe-Ti oxides, but it still is moderately to strongly incompatible. We now state this explicitly in **line 125**.

line 74: One of the reasons you stated above for dealing with melt incs is because of pre-eruptive degassing. If that was so, then why would you need to correct?

In the new Methods section, we emphasize that in order to determine the quantities of Li available from eruption of rhyolites, it is necessary to analyze melt inclusions to account for the major losses of Li expected due to vapor exsolution on eruption and subsequent alteration of rocks in the weathering environment. But it is also possible that prior to eruption, the magmas reached vapor saturation at depth, resulting in some Li loss by this degassing. In the vapor correction procedure we assess whether there was any vapor lost during the interval that melt inclusions were being trapped in quartz, and, if so, we calculate its expected effect on the abundance of Li in order to arrive at vapor-loss-corrected magmatic concentrations of Li.

line 118: What is “it” here?

We rewrote this sentence to remove this ambiguity.

line 124: Also Sr, Nd isotopic or others?

We added “for Sr and Nd” to make it obvious that we are referring to those isotopic systems (**line 195**).

line 126: You mean enriched in Rb relative to Sr, or having a high Rb/Sr, right?

Yes; reworded to clarify this.

line 136: This concentration is suspiciously like that reported earlier by London et al. (1989).

We added the London et al. reference as well as a reference to MacDonald and Bailey (1992) (**line 223**).

Reviewer #2 (Remarks to the Author):

January 7, 2017

Review of NCOMMS-16-29348

Adam Simon

This paper presents and interprets new data for the enrichment of Li in rhyolitic magmas, and stresses the importance of such rocks as a future resource of Li. As Li is considered a critical metal necessary for production of Li batteries, and worldwide consumption of Li is expected to increase in the next decades, the paper is appropriate for a wide audience. It is not stressed much in the paper, but it is incredibly important to diversify the global Li supply chain, which at present is dominated by a very small number of countries. The situation is not as dire as it is for rare earth metals (Chinese monopoly), but most of the EU and the United States are almost entirely reliant on Li imports. For example, the United States imports almost two-thirds of consumed Li. For most countries in the EU, the import reliance is higher. Thus, identifying 'new' resources of Li that are likely economically viable from a mining perspective is important. Overall, the paper is easy to read and the interpretations are rooted in the data presented. My comments below are based on reading the paper in sequence. If I make a comment that is subsequently addressed in the paper, please forgive that.

As noted above, we have rewritten the abstract and introduction to emphasize the economic and strategic importance of identifying new Li resources, from perspectives of supply/demand and diversifying the global Li supply.

L26: The concept of "resource" is likely unknown to readers of Nature Communications. I suggest to define this somehow in the space allowed. Ideally, it would be great to present the economic reserves for Li for each type of resource to give the reader a sense of how price plays a role in making any particular resource economical to extract.

We now define both 'reserve' and 'resource' (lines 29 – 38), and, as noted above, include a new figure (Figure 1) that shows the global resources and reserves by deposit type and country to highlight the strategic necessity of exploring for additional Li sources.

This is a huge range for Li resources. I checked the USGS December 2016 annual report for Li resources and they quote 34 million tonnes total worldwide. Where does the upper estimate of 65 million tonnes originate?

The upper estimate is from Vikstrom et al. (2013). We reworded the introduction to focus on the resource and reserve data from the latest USGS commodity report and Christmann et al. (2015), both of which estimate the global Li resource to be ~41-45 Mt. (7 Mt in U.S. plus 34 Mt elsewhere).

Perhaps something else would be to report the reserves / production for each of the resources to give the reader perspective on the expected depletion rate of a particular resource. Right now brines are cheap, but they'll be exhausted in the near future as the Gigafactory and others around the world produce more Li batteries. Also, Li sources other than brines are becoming economically competitive. I know the authors are space limited, but it would be great to add a

sentence or few (or a table) that shows the reader the expected lifetime of each Li resources relative to forecasted consumption. Those are available via the USGS annual reports and other free sources. Most people simply assume an inexhaustible source of metals, but there are limits. And as one source becomes exhausted, others become economical.

This is the main selling point to a wider audience who (like it or not) do not care about how or why the rhyolitic magmas are enriched in Li. The mining community cares if it helps exploration strategies. The USGS certainly cares as they are charged (among other things) to identify reserves of strategic/critical metals. The business community (Tesla) cares only that sustainable Li resources exist to build their battery. So, some of my comments (including the one above) are meant to help the paper out of the petrology silo.

As noted previously, we have modified the abstract and introduction, consistent with the reviewers comments above, to provide the economic and political context that makes the current study important.

L40-41: In 2015, this was not true. The spodumene Li mining operations in Australia gained market share and were the second largest producer of Li to global markets. In fact, more Li was mined from spodumene that was mined from the salars in Chile (note that production from Bolivia continues to have zero supply to global markets). You should revise your statement here as market forces are reshaping the economics of the Li supply chain.

We now state this explicitly (lines 33 – 36).

L43-44: The writing is confusing. Granite and rhyolite are essentially compositionally identical. What is different about a rhyolitic magma chamber and a granitic magma chamber? As written, a separate magmatic volatile phase percolates and accumulates in the top of a rhyolitic magma chamber, but not a granitic chamber. And likewise, bubbles only accumulate at the top of a rhyolitic magma chamber, but not a granitic magma chamber. Is this true? I have seen lots of granitic plutons that experienced volatile exsolution and loss of a magmatic volatile phase to the surrounding rock.

We have rewritten this section to remove the confusing wording re granite/rhyolite and to emphasize the main point: Li is enriched in the most evolved magmas (lines 53 – 56).

L62: Does “rhyolitic magma” equal “rhyolitic whole rock”, where you assume that the whole rock represents the magma that existed prior to eruption? That is, there was no mass loss to a vapor phase prior to and/or during eruption? Perhaps it is appropriate to use “whole rock” as that is what you measured.

We have rewritten this sentence (line 109) to make it clear that these are melt inclusion concentrations, not whole rock or magma concentrations. As such, they represent the concentration in the melt phase at the time of entrapment, which is what we are after.

L75: Is the partition coefficient for Li static, or a function of melt composition, which, in turn, at a given pressure and temperature controls the composition of the vapor phase? In this case, the

total Cl concentration of the vapor phase, which is important for scavenging Li from the melt. Do you have any independent (i.e., fluid inclusion) evidence for the salinity of the vapor phase in your samples? Are there any co-entrapped fluid and melt inclusions in the quartz phenocrysts?

We did not observe fluid inclusions in the quartz crystals, so have no direct measure of salinity of any coexisting vapor phase.

The vapor/melt partition coefficient for Li, is, of course, a function of temperature, pressure, and melt composition. We used the most relevant values available, those derived from experiments by Webster et al. (1989) and Webster and Holloway (1990). In Supplementary Figure S5, we show the effect of varying the Li partition coefficient on the vapor-loss correction.

L77: Unless apatite is part of the crystallizing assemblage, whereupon the F/Cl of the melt will decrease with fractionation. So this assumes apatite is not a stable phase. Was apatite absent during fractionation of all the systems?

Apatite is a crystallizing phase in the rhyolitic systems we studied; however, it is such a small proportion of the fractionating assemblage (< 1 vol. %) that the bulk distribution coefficients for F and Cl are much less than unity. We now state this explicitly in the text (lines 416 – 423).

L94: Was the oxidation state of all the systems similar? This is important when comparing potential loss of sulfur to a volatile phase, a process that is strongly redox dependent. A slight shift of oxidation state across the sulfide – sulfate transition will decrease the vapor/melt partition coefficient by at least one order of magnitude.

Apatite can also incorporate sulfur, and this is highly dependent on the oxidation state of the system (see new paper by Konecke et al., 2017, American Mineralogist: Co-variability of S⁶⁺, S⁴⁺, and S²⁻ in apatite as a function of oxidation state: Implications for a new oxybarometer).

The oxidation state of all the systems we studied was near the FMQ buffer, so there is no reason to believe that fO₂-driven changes in the partitioning of S into apatite would have a large effect on the F and Cl abundances in apatite. And, as stated above, the amount of apatite in the crystallizing assemblage is too small in any case to affect the melt F/Cl ratio substantially. We removed the brief mention of S concentrations from the text to avoid any confusion relating to S oxidation.

L 101: Should be singular tense as a magma cannot have more than one concentration of Zr (unless you refer to the concentration of Li in discrete phases of the magma).

We corrected this.

L131-133: I'm not sure I entirely follow the first part of the sentence. Do you mean that you measured Li in the Hideaway Park whole rock by solution chemistry and also by fusion LA-ICPMS of a glass bead? Why the half order of magnitude range of Li concentrations? And one order of magnitude variability for Spor Mountain samples?

We rewrote this section to make it clear that we are simply reporting the LA-ICPMS analyses of melt inclusions by Mercer et al. (2015), who provided us with an aliquot of their Hideaway Park sample. Because we do not have access to SEM imagery of the samples analyzed by Mercer et al. (2015), we cannot speak to whether the variability they report is geological or a function of incomplete homogenization—something they noted in their samples. We added some simple diffusion calculations to hypothesize that their low and variable Li concentrations may be a function of longer homogenization experiments (**lines 113 – 121**), which resulted in loss of Li from the melt inclusions to the host quartz. We added a few words to indicate that the variability in the Spor Mtn samples is a function of the analyses being nonhomogenized and therefore likely have lost some Li to vapor bubbles (**line 219**).

L135: As written, some analyses of Hideaway Park have higher Li concentrations than Macusani, but other samples have lower concentrations. So this statement is not strictly true. What is the average and standard deviation of the Hideaway Park samples? Perhaps the concentrations of the two locations are statistically similar?

We rewrote this to make it clear that we are referring to volcanic glass that degassed during eruption (**line 222**).

L136: Considering that these are tin granites, which are known to evolve at reduced fO_2 , relative to typical arc magma systems, is there anything about the oxidation state of the systems that may have played a role in controlling the fractionating assemblage that, in turn, led to Li enrichment?

The oxidation state of all analyzed systems was at about FMQ given their phenocryst assemblages, with slightly higher values in Hideaway Park (though still below NNO)(Mercer et al., 2015). Because of this, we do not think our measured wide Li variations are a function of fO_2 . Li is strongly incompatible in all major phenocrysts in rhyolitic magmas with the exception of biotite, in which it is weakly compatible. While differences in magmatic fO_2 can affect the composition and the abundance of biotite found in rhyolitic magmas, the low abundance of biotite in the crystallizing assemblages in these evolved rocks combined with the weak partitioning of Li into biotite results in bulk distribution coefficients much less than unity. Along these lines, Icenhower and London (1993, 1995) showed experimentally that crystallization of biotite is not accompanied by depletion of Li in residual magma. Moreover, among the samples studies, the Hideaway Park Tuff is the only one to contain biotite, yet it has the melt inclusions with the the highest measured Li concentrations. We have modified the text to clarify this issue (**lines 126 – 131**).

(The low fO_2 and elevated Li in tin granites and Macusani rhyolite are not indicative of a concentration process controlled by fO_2 ; both characteristics are merely reflecting the very large pelitic sedimentary component in the source regions of the magmas.)

L146: But this depends on the total tonnage of the resource host rocks, and not simply the concentration of Li.

Yes; here we are just discussing the total amount of erupted Li. We refined this section to make our calculations and assumptions more explicit (**lines 235 - 242**).

L154: Lavas of any composition? Or evolved, rhyolitic lavas?

We added “rhyolitic” as post-caldera basalts have low Li concentrations and are not providing significant additional Li to the system (**line 251**).

L155: “...a subjacent shallow magma chamber”. There is a lot of discussion among volcanologists and folks in economic geology as to whether the addition of such volatiles from deeper levels of the volcanic plumbing system are in fact required to drive eruption of the magma that results in caldera formation.

We reworded this part of the sentence to make our point more clear (**line 252**).

L170: Is this accurate? As written, a rhyolitic magma physically assimilates a lithospheric melt. Do you perhaps mean that the rhyolitic magma itself is the product of lithospheric melting? Revise to readability.

We rewrote this to be clear that the Li-enriched rhyolite magmas could assimilate the crust or be formed by partial melting of the crust (**lines 272 – 273**).

Supplement

I know and respect Paul Wallace, but he is far from the first person to report that rehomogenization of melt inclusions is necessary. Folks in economic geology spent years convincing volcanologists that the vapor bubble is not a vacuum. The work of Rosario Esposito and Bob Bodnar is perhaps the most convincing over the past few years (Esposito et al., 2014. An assessment of the reliability of melt inclusions as recorders of the pre-eruptive volatile content of magmas, *Am Min*). It is also possible to ablate the entire inclusion without rehomogenization and obtain accurate compositions of the inclusion. Especially when quartz is the host and Li is incompatible in quartz, this should be straightforward. Only if you are analyzing a highly volatile fugitive component of the inclusion (e.g., CO₂, CH₄, S) is it critical to homogenize. But I am used to doing this with a larger diameter laser ablation ICP-MS where it is easy to ablate the inclusion and a small volume of host material and deconvolve the mixed signal. So it is not relevant for your analytical technique.

We added the Esposito et al. (2014) reference to the homogenization discussion in the Methods section (**line 284**). Like Reviewer #2 mentions, the LA-ICP-MS technique is useful but not relevant for this paper, which uses *in situ* SIMS analyses.

Did you do any analytical traverses away from a rehomogenized inclusion away into the quartz host to assess diffusive loss of Li or other elements either prior to sampling or during rehomogenization?

We did not perform analytical traverses away from the melt inclusions because we limited our homogenization runs to ~55 minutes to minimize diffusion of Li into the host quartz, following methods of Hofstra et al. (2013), based on existing data on Li diffusion in quartz (~3x10⁻¹¹ m²/s; Charlier et al., 2012). We added a few sentences to this effect (**lines 115 – 121**).

Did you work on melt inclusion assemblages? How did you assess that the inclusions were trapped at the same stage of quartz growth?

Few of the quartz phenocrysts in this study contained multiple inclusions exposed on the analyzed surface. In Figure 2 we show a CL image of one that does; the inner melt inclusion has lower concentrations of incompatible trace elements than the outer inclusion.

There is no way to assess whether inclusions were trapped at the same stage of quartz growth in different samples. This isn't necessary to demonstrate how Li behaves on fractionation, because we arrive at this conclusion by comparing the abundance of Li versus other incompatible elements like Rb in all the analyzed inclusions in a given sample. They show positive correlations, indicating that Li is enriched in the melt by crystallization. The fact that the linear trends differ in the samples studied indicates that systems begin at different Li "baselines"; it is the baseline.

In Figure S3, what are the 3 inclusions that you annotate with "trapped other magma"? Are those inclusions spatially associated with other inclusions for which data are presented? Do you imply that the quartz host for some inclusions was recycled during magma mixing from another magma?

We removed this figure from Supplementary Material because it includes a sample not reported from this study, and because we think the aberrant analyses may have been due to quartz overlap by the ion beam, a problem we discuss in the Methods section.

When you refer to post-entrapment degassing and loss of Li, S, Cu, technically, it is the melt phase that reaches volatile saturation and degasses.

Yes; we reworded to correct this.

Figure S4. Another semantics issue, but I would not say "elements volatilized on eruption". The melt reaches volatile saturation and those volatile-compatible elements are partitioned from the melt to the volatile phase, which is the driver for the eruption (i.e., the mass transfer of volatiles from melt to fluid occurs prior to eruption). Yes, some additional loss occurs during eruptive decompression, but much occurs prior to eruption.

(Now Figure S3). We corrected this; the figure now labels these elements as "volatilized elements."

Are the partition coefficient values you assume appropriate for your bulk compositions? The partitioning of Cl is highly sensitive to changing melt composition, temperature and pressure. Also, the acidity of the exsolved vapor is strongly dependent on the alkalinity of the silicate melt and, in turn, can affect Li partitioning.

We used literature values for the partition coefficients for F, Cl, and Li that are most appropriate for the compositions of the analyzed samples, largely from the work of Jim Webster. Interested readers can easily adjust these coefficients (and thus the vapor correction factor) to systems with

different conditions using the equations provided in the Methods section. We include a schematic graphic in the supplemental material to show how varying the partition coefficients will change the vapor-loss correction.

Is apatite never present in any of the systems? Apatite is seemingly ubiquitous in intermediate to felsic magmas. Further, the partitioning of F and Cl between silicate melt and apatite depends on melt composition, pressure and temperature. I am not intimately familiar with the rocks in this study, but find it surprising that apatite is not present.

As discussed above, apatite is present in the samples studied but makes up such a small proportion of the crystallizing assemblage, that the bulk distribution coefficients for F and Cl are less than unity; as such, crystallization of apatite will not have a discernable effect on the F/Cl ratio. We adjusted the text and Methods section make this explicit (**lines 416 – 423**).

Reviewer #3 (Remarks to the Author):

This interesting manuscript provides important new data on lithium in volcanic systems that better support modeling of the generation of lacustrine-based Li deposits located in volcanic calderas. The melt inclusion data are of high quality and are convincingly applied to Li deposit formation.

This manuscript is worthy of publication after minor revision as detailed below.

Major comments:

I. The abstract could be rewritten to better relate the primary data source of this study (i.e., melt inclusions) to the larger processes of Li concentration in sedimentary environments. For example, it does not logically follow (for non-geological background readers) that measurements of Li in silicate melt inclusions provide direct constraints on Li concentrations/deposits in intracontinental settings, see lines 15-18, given that other processes (weathering of volcanic materials and the precipitation of Li-enriched clay minerals) are involved. Folks with geo-backgrounds will be able to understand this, but given the importance of Li-bearing batteries (and current interest in related energy storage issues), presumably other readers will be interested in this paper when published so it will be useful to expend extra effort to make the abstract more broadly understandable in this context.

We rewrote the abstract per Reviewer #3's suggestion to emphasize the conclusions of the paper in the context of exploration for new lithium resources.

II. In the Supplementary Information 1 document, the authors apply (F/Cl) ratios to constrain the likelihood and quantity of magmatic vapors in the systems investigated. The slopes of (F/Cl) plots are interpreted in this regard. This approach assumes that fractional crystallization (or other potential processes such as magma mixing) has no measurable influence on the behaviors of F, Cl, and Li at the stage of vapor saturation (specifically, that Li, Cl, and F increase similarly with fractional crystallization and/or that F is not fractionated relative to Cl by any process other than

loss to fluid). This does not, necessarily, seem to be a reasonable assumption for rhyolitic systems, and should be better justified/explained in the revised manuscript.

Also, to this point, it would be useful for the authors to include any previously published constraints on the likelihood and timing (and quantities, if estimated) of a vapor phase in these magmatic systems studied, if the vapor phase was previously proposed/determined to exist - for the revised manuscript – given the importance of a magmatic vapor phase in Li transport.

We think it is a reasonable assumption that Li, F, and Cl increase similarly during fractional crystallization because all three elements form positive linear trends with the incompatible element Rb (and Zr for peralkaline magmas), indicating that they are behaving incompatibly and consistently throughout the evolution of all systems analyzed at the time of quartz crystallization. We added a few sentences to the text and Methods section to clarify this (**lines 411 - 414**).

Minor, annotated comments in the text pdf, by line numbers:

1. 37-39: spodumene is not a water-rich mineral; text should be corrected.

Corrected by removing “water-rich” from this sentence (**lines 40 – 41**).

2. 41-44: given the prior statement about Li concentrating in lepidolite and spodumene, it is important to distinguish crystallization of the dominant (non-rare) silicate minerals such as feldspar, quartz, non-Li bearing micas, etc. (for the non-geology reader) – otherwise this might appear contradictory. For example, this is better explained in lines 69-70.

Reworded to indicate that we are referring to the major phenocrysts crystallizing in rhyolites (**line 55**).

3. 62-66: again for the non-geo reader, please specify explicitly if the Li contents of rhyolitic magmas are melt inclusion data (not whole-rock data). Also, melt inclusions provide constraints on melt compositions and not constraints for phenocryst-bearing magma compositions, to be specific, since the magma represents melt and crystals. Similarly, in the supplementary table that describes the various volcanic fields (and rock sources), please specify that the average Li contents determined in this study are from melt inclusion analyses.

We now state clearly that these are melt inclusion concentrations and that they reflect the composition of the melt, not the magma (**line 109**). We have also made this more clear by including measured values in the new Table 1.

4. 143: sample 90-5.1.1 of Webster et al. (1996) contains 0.33 wt% Li on average (see Table 4).

We are reporting the average of all five lavas in this table, not individual samples. We have reworded this to make this explicit, and now report the range in average Li concentrations from the lavas in this study. We also correct the error (average is 1,400 ppm, not 2,400 ppm) (**lines 230 – 231**).

5. Other: In the Supplementary Information 1 document, the results of rehomogenization of the melt inclusions are described texturally, but please explicitly state whether, or not, only bubble-free and crystal-free melt inclusions were analyzed for this study.

We now state this explicitly in the Methods section (lines 376 – 377).

6. Other: In the supplementary table that describes the various volcanic fields (and rock sources), it would be very useful/helpful if some basic compositional information was included for row 4 (sample types)... brief information on alkalinity/aluminosity should be added.

We added classification (all rhyolite) and alkalinity (peralkaline vs metaluminous vs peraluminous) information to row 4.

7. The figure axis labels for Figures 2 and 3 should explicitly state “Li (ppm) in melt inclusions” and “Rb (ppm) in melt inclusions”, “Nd epsilon in whole rocks”, etc.

We updated the axis labels in Figures 2 and 3 per this suggestion.

8. Does Figure 4 represent a generic/general model as the figure implies or does it specifically represent the Kings Valley Li deposit of the McDermitt caldera (as text lines 158-159 state)? I ask this because the diagram includes a porphyritic intrusion and I am curious to know if this intrusion is broadly applicable to caldera Li deposits in general or it just applies to the Kings Valley deposit/caldera?

We added a sentence to indicate that although this model is based on the McDermitt Caldera, it can be applied broadly to other calderas. The intrusion along the caldera ring fracture is hypothetical, but is consistent with eruption of rhyolitic lavas and the development of hydrothermal systems along ring fractures in most caldera systems (lines 267 - 269).

Jim Webster

Reviewers' Comments:

Reviewer #1:

Remarks to the Author:

The authors dealt productively with my comments and criticisms, in particular addressing the question I had about the correction for Li lost during degassing. I maintain, having re-read the pertinent papers by Pehlken, etc., that the predictions of grossly increased Li demand in the coming half century are speculative, with many estimates of decreased global demand compared to current. Yes, I understand that geopolitically nations hate to have all their natural resource eggs in too few baskets, but that is the case for plenty of substances—I don't particularly see this as the reason that puts this manuscript over the top for becoming a Nature contribution. Nonetheless, I do think the work stands well on its own and would attract attention if published here.

Reviewer #2:

Remarks to the Author:

I wrote a detailed review for this manuscript and feel that the authors have done a good job addressing each substantive and stylistic comment. Here are a couple of small things that will improve the readability of the manuscript.

L14 should read: "...formed from weathering of Li-enriched magma..."

L54 should read: "...crystallize from rhyolitic melt..." considering that magma = melt + crystals +/- volatile phase and minerals crystallize from melt. I know this may seem trivial, but it is an important distinction.

I think this manuscript is appropriate for publication in Nature Communications and will be well received by the community.

Adam Simon

Reviewer #3:

Remarks to the Author:

The revised manuscript is much improved, following on the comments and suggestions of each of the three reviews.

I full support publication of this version of the manuscript.

I offer two minor typographical issues:

1. Line 311: the sentence is missing the second, closing ")."
2. Lines 376-377: The new text "All analyses reported in this study are vapor bubble- and crystal-free." Should be modified to something like "All melt inclusion glasses analysed in this study are vapor bubble- and crystal-free"

Jim Webster

Reviewer #1 (Remarks to the Author):

The authors dealt productively with my comments and criticisms, in particular addressing the question I had about the correction for Li lost during degassing. I maintain, having re-read the pertinent papers by Pehlken, etc., that the predictions of grossly increased Li demand in the coming half century are speculative, with many estimates of decreased global demand compared to current. Yes, I understand that geopolitically nations hate to have all their natural resource eggs in too few baskets, but that is the case for plenty of substances—I don't particularly see this as the reason that puts this manuscript over the top for becoming a Nature contribution. Nonetheless, I do think the work stands well on its own and would attract attention if published here.

Reviewer #2 (Remarks to the Author):

I wrote a detailed review for this manuscript and feel that the authors have done a good job addressing each substantive and stylistic comment. Here are a couple of small things that will improve the readability of the manuscript.

L14 should read: "...formed from weathering of Li-enriched magma..."

We rewrote this to read: "...formed on eruption and weathering of Li-enriched magma ..."

L54 should read: "...crystallize from rhyolitic melt..." considering that magma = melt + crystals +/- volatile phase and minerals crystallize from melt. I know this may seem trivial, but it is an important distinction.

We corrected this.

I think this manuscript is appropriate for publication in Nature Communications and will be well received by the community.

Adam Simon

Reviewer #3 (Remarks to the Author):

The revised manuscript is much improved, following on the comments and suggestions of each

of the three reviews.

I full support publication of this version of the manuscript.

I offer two minor typographical issues:

1. Line 311: the sentence is missing the second, closing ")".

We corrected this (now line 306).

2. Lines 376-377: The new text “All analyses reported in this study are vapor bubble- and crystal-free.” Should be modified to something like “All melt inclusion glasses analysed in this study are vapor bubble- and crystal-free”

We corrected this (now lines 403 - 404).

Jim Webster